# Integrated insights into gut microbiota and metabolomic landscape in breast cancer patients undergoing adjuvant endocrine therapy

Taha Majid Mahmood Sheikh,[1] Fen Yao,[2] Zhenyan Liu,[3] Muhammad Shafiq,[2] Jilong Wu,[1] Areeba Khalid,[4] Qingdong Xie,[1] Xiaoyang Jiao,[1] Weitao Shen[3]

**ABSTRACT**  Gut microbiota and systemic metabolites are critical for breast cancer progression and therapeutic response. This study investigated gut microbiota and metabolic profiles of breast cancer patients before and after adjuvant endocrine therapy (AET). Using 16S rRNA sequencing and untargeted metabolomics, we identified significant disruptions in microbial diversity and metabolic pathways. Alpha diversity was reduced in pre-AET patients, with partial restoration observed post-AET. Key genera, such as *Bifidobacterium* and *Coprococcus*, were enriched in pre-AET patients, whereas *Proteus* and *Methylobacterium* were enriched in post-AET patients. Metabolomic analysis revealed significant reductions in the levels of vitamin B6 metabolites in both the pre-AET and post-AET groups compared to those in the healthy control (HC) group, indicating potential nutrient deficiencies or metabolic stress. Elevated cholesterol and estrogen metabolite levels in pre-AET patients reflect dysregulated lipid and hormone metabolism, with post-AET decreases in estrogen metabolites, confirming therapeutic efficacy. Correlation analysis revealed that *Klebsiella_724518* was positively correlated with estrogen and vitamin B6 metabolites, whereas *Proteus*, *Methylobacterium*, *Treponema_D*, and *Holdemanella* were negatively correlated with cholesterol. Receiver operating characteristic (ROC) curve analysis identified estriol (area under the curve [AUC] = 1.000) as a strong diagnostic biomarker for distinguishing HCs from pre-AET patients, whereas cholesterol (AUC = 0.880) and estradiol-17β (AUC = 0.870) were highly effective in monitoring the therapeutic response to AET. This study highlights the role of gut microbiota and its metabolic byproducts in breast cancer development and treatment outcomes. It also reveals promising microbial and metabolite signatures for non-invasive cancer detection, tracking progression, and monitoring treatment response.

**IMPORTANCE**  Breast cancer progression and treatment response remain challenging to predict and monitor effectively. Our findings demonstrate the dual role of the gut microbiota and its metabolic products in influencing both. Strong correlations between specific microbial taxa and key metabolites provide new mechanistic insights into the influence of gut microbes on therapeutic outcomes during endocrine therapy. Importantly, we identified high-performance biomarkers, with estriol showing perfect diagnostic accuracy (AUC = 1.000) and cholesterol effectively monitoring treatment response (AUC = 0.880), highlighting their potential for non-invasive clinical applications. This study provides a foundation for applying gut microbiome research to develop clinical tools that could improve breast cancer management.

**KEYWORDS**    breast cancer, adjuvant endocrine therapy, gut microbiota, metabolomics, vitamin B6 metabolism, steroid hormone biosynthesis, microbial biomarkers

**Peer Reviewer** Abid Ullah Shah, Nanjing Agricultural University, Nanjing, Jiangsu, China

Address correspondence to Xiaoyang Jiao, xyjiao@stu.edu.cn, or Weitao Shen, 17f2wtshen@stu.edu.cn.

Taha Majid Mahmood Sheikh, Fen Yao, and Zhenyan Liu contributed equally to this article. Authors order was determined randomly.

The authors declare no conflict of interest.

See the funding table on p. 17.

Breast cancer is the most common cancer in women worldwide, with an estimated 2.3 million cases and 680,000 deaths reported in 2020 (1), and in 2025, approximately 316,950 women will be diagnosed with invasive breast cancer (2). It remains one of the leading causes of cancer-related deaths in women (3, 4). In China, the incidence of breast cancer has been rapidly increasing, contributing to a significant disease and economic burden (5). Estrogen, a key hormone in females, is a significant factor in the development and progression of breast cancer (6, 7). Consequently, adjuvant endocrine therapy (AET), which targets the effect of estrogen, continues to be a critical strategy for eradicating and controlling estrogen-dependent breast cancer (8, 9). Approximately 70% of women with breast cancer express estrogen receptors (ER), progesterone receptors (PgR), or both, classifying them as hormone receptor-positive (HR+) (10). AET is highly effective and suitable for most women with ER-positive or PR-positive breast cancer (11–13). Therefore, it is the most widely used treatment for patients with breast cancer globally, significantly reducing recurrence rates and improving overall survival outcomes (14, 15).

The gut microbiota plays a critical role in the regulation of estrogen metabolism, influencing systemic estrogen levels through the enterohepatic circulation (16, 17). The liver secretes estrogens into the bile, which are then reabsorbed in the intestines. The gut microbiota plays an essential role in this process by deconjugating estrogen metabolites using enzymes such as β-glucuronidases, converting them back into their active forms, and facilitating their reabsorption into the bloodstream (18). Through this intricate interaction, the gut microbiota can regulate estrogen levels, which is crucial in hormone-dependent diseases such as breast cancer.

Recent studies suggest that AET may modulate the gut microbial composition and function. For example, aromatase inhibitors have been associated with shifts in microbial diversity and alterations in bacterial taxa linked to estrogen metabolism (19). Additionally, tamoxifen treatment has been shown to influence bile acid metabolism, which is closely associated with microbial activity (20). There is limited understanding of how different AET regimens affect gut microbial diversity and the production of metabolites related to hormone regulation. This knowledge gap is critical because microbe-derived metabolites may serve as biomarkers for treatment efficacy or be leveraged to improve therapeutic outcomes.

In addition to regulating estrogen, gut microbiota produces various microbial metabolites that influence systemic metabolism and immune function, further affecting cancer progression. Short-chain fatty acids (SCFAs), such as acetate, propionate, and butyrate, are produced during the fermentation of dietary fiber by gut bacteria. These metabolites can induce apoptosis in cancer cells through oxidative stress and may reduce the invasive potential of breast cancer cells (21, 22). Similarly, lithocholic acid, a secondary bile acid produced by the gut microbiota, inhibits breast cancer cell proliferation and invasion by activating TGR5 receptors (23).

While the relationship between gut microbiota and breast cancer has been explored, the functional contribution of microbial metabolites to breast cancer progression remains unclear. Microbial metabolites act as biochemical converters that link the gut microbiota to various metabolic pathways involved in cancer development (24–26). However, the characterization of microbial metabolites in gut samples and their role in breast cancer progression remains an emerging area of research. Notably, there is limited understanding of how AET influences gut microbial diversity and the production of specific metabolites associated with hormonal regulation, which could be leveraged to improve therapeutic outcomes.

This study explores the impact of AET on gut microbiota and metabolite profiles in breast cancer patients. Specifically, we sought to characterize microbial shifts in response to therapy and understand the potential role of gut microbiota and metabolites as biomarkers for treatment response. By integrating 16S rRNA gene sequencing and non-targeted metabolomics, we gained insights into the influence of AET on microbial diversity and metabolic pathways. Additionally, this study investigated the crosstalk

between the gut microbiota and metabolic pathways, particularly those related to hormone metabolism and lipid regulation, providing new insights into the potential of microbiota as biomarkers for therapy monitoring and personalized treatment.

## MATERIALS AND METHODS

### Study participants and inclusion/exclusion criteria

This study enrolled ten postmenopausal women with hormone receptor (HR)-positive breast cancer who were pathologically diagnosed by the Department of General Surgery at The Second Affiliated Hospital of Shantou University, Guangdong Province, China. All participants were newly diagnosed with breast cancer and scheduled to receive adjuvant endocrine therapy (AET). The study also included a control group of ten healthy individuals matched to the breast cancer patients based on age and sex. There was no significant age difference between the two groups ($P = 0.115$) as mentioned in Table 1. Notably, all patients and the control group in this study were from the same geographic area, and they share similar lifestyle factors and have similar dietary habits. Moreover, adherence to AET was closely monitored throughout the study period. Follow-up assessments confirmed that the patients exhibited high medication compliance during the treatment.

Participants were excluded if they met any of the following criteria: (i) a history of inflammatory bowel disease (IBD), intestinal tumors, or other infectious diseases; (ii) use of probiotics, prebiotics (e.g., yogurt), broad-spectrum antibiotics, steroid hormones, or Chinese herbal medicine within three months prior to stool sample collection; and (iii) a history of blood transfusion within six months prior to blood collection. All participants provided informed consent, and the study was approved by the Ethics Committee of Shantou University Medical College and the Second Affiliated Hospital of Shantou University Medical College.

### Sample collection and storage

Stool samples (approximately 2 g) were collected from each breast cancer patient at the time of initial admission (pre-AET) and six months after the initiation of the adjuvant endocrine therapy drugs (post-AET). These drugs included aromatase inhibitors (letrozole or exemestane) and a selective estrogen receptor modulator (tamoxifen). All these drugs target the estrogen receptor pathway in hormone receptor-positive breast cancer. These drugs were analyzed as one group in this study, as the focus was on the general impact of AET on gut microbiota and metabolomics. Fasting venous blood (2 mL) was drawn in the morning at both time points, and serum was isolated by centrifugation at 3000 rpm for 10 minutes. Stool samples (approximately 2 g) and 2 mL of fasting venous blood were collected from each healthy control participant under the same conditions. All stool and serum samples were stored at −80°C until further analysis.

### Biochemical and hormonal assays

Serum levels of estradiol (E2), triglycerides (TG), total cholesterol (TCH), high-density lipoprotein (HDL), low-density lipoprotein (LDL), apolipoproteins A1 (APO-A1) and B (APO-B), alanine aminotransferase (ALT), and aspartate aminotransferase (AST) were measured in patients with breast cancer before and after AET and in the healthy control group (HC). Estradiol (E2) levels were quantified using a Siemens DXI800 automatic chemiluminometer. Biochemical parameters (TG, TCH, HDL, LDL, APO-A1, APO-B, ALT, and AST) were measured using a Beckman AU5800 automatic biochemical analyzer. All assays were conducted using the original matching kits.

### DNA extraction and library construction

Genomic DNA was extracted from fecal samples using the CTAB method, and its purity and concentration were evaluated using 1% agarose gel electrophoresis. The DNA

**TABLE 1** Clinical characteristics and laboratory parameters of the patients and control[a]

| Indexes | pre-AET (n = 10) | post-AET (n = 10) | HC (n = 10) | P-value |
|---|---|---|---|---|
| Age | 58.80 ± 6.65 | 58.80 ± 6.65 | 54.1 ± 2.38 | 0.115 |
| Gender (female) | 10 (100%) | 10 (100%) | 10 (100%) | |
| Clinical stage | | | | |
| IIA | 5 (50%) | | | |
| IIA~IIIA | 1 (10%) | | | |
| IIIA | 2 (20%) | | | |
| IIIC | 2 (20%) | | | |
| HER-2 | | | | |
| − | 2 (20%) | | | |
| + | 3 (30%) | | | |
| ++ | 3 (30%) | | | |
| +++ | 2 (20%) | | | |
| FISH(HER-2) | | | | |
| Amplified | 1 (10%) | | | |
| Not amplified | 9 (90%） | | | |
| Tumor stage | | | | |
| T1 | 3 (30%) | | | |
| T2 | 6 (60%) | | | |
| T3 | 1 (10%) | | | |
| Node stage | | | | |
| N0 | 2 (20%) | | | |
| N1 | 3 (30%) | | | |
| N1-N2 | 1 (10%) | | | |
| N2 | 2 (20%) | | | |
| N3 | 2 (20%) | | | |
| Molecular type | | | | |
| Luminal A | 1 (10%) | 1 (10%) | | |
| Luminal B | 9 (90%) | 9 (90%) | | |
| Histological type | | | | |
| Invasive ductal carcinoma | 9 (90%) | 9 (90%) | | |
| Encapsulated papillary carcinoma | 1 (10%) | 1 (10%) | | |
| Medicine | | | | |
| Aromatase inhibitors (letrozole, exemestane) | | 9 (90%) | | |
| Tamoxifen | | 1 (10%) | | |
| TG | 1.56 ± 0.96 | 1.57 ± 0.48 | 1.46 ± 0.91 | 0.943 |
| TCH | 5.91 ± 1.27 | 5.25 ± 1.05 | 5.86 ± 1.89 | 0.530 |
| HDL | 1.53 ± 0.24 | 1.37 ± 0.28 | 1.26 ± 0.42 | 0.181 |
| LDL | 3.79 ± 0.92 | 3.21 ± 0.77 | 4.04 ± 1.65 | 0.287 |
| APOA1 | 1.54 ± 0.16 | 1.56 ± 0.24 | 1.63 ± 0.24 | 0.647 |
| APOB | 1.15 ± 0.28 | 1.00 ± 0.27 | 1.06 ± 0.27 | 0.446 |
| ALT | 22.30 ± 12.29 | 30.00 ± 21.59 | 10.80 ± 6.46 | 0.025 |
| AST | 25.00 ± 4.99 | 26.70 ± 6.5 | 19.00 ± 7.9 | 0.035 |
| GGT | 35.90 ± 19.44 | 57.30 ± 57.98 | 33.50 ± 25.4 | 0.324 |
| CG | 2.07 ± 1.46 | 1.64 ± 2.14 | 0.74 ± 0.32 | 0.149 |
| E2 | 13.22 ± 9.33 | 0.98 ± 0.97 | 10.15 ± 3.63 | 0.000 |

[a]pre-AET, pre-adjuvant endocrine therapy for breast cancer; post-AET, post-adjuvant endocrine therapy for breast cancer; HC, healthy control; P, P value; HER-2, human epidermal growth factor receptor 2; C-erBb-2, C-erBb-2 oncogene; TG, triglyceride; TCH, total cholesterol; HDL, high-density lipoprotein; LDL, low-density lipoprotein; APOA1, apolipoprotein A1; APOB, apolipoprotein B; ALT, alanine aminotransferase; AST, aspartate transaminase; GGT, gamma-glutamyltransferase; CG, cholyglycine; E2, estradiol.

was then diluted to a concentration of 1 ng/µL. PCR amplification was performed on the V3-V4 region of the 16S rDNA gene using the primers 341F CCTAYGGGRBGCASCAG (forward primer) and 806R GGACTACNNGGGTATCTAAT (reverse primer). The PCR products were purified using 1% TAE agarose gel electrophoresis. DNA was recovered using the Universal DNA Purification and Recovery Kit (TianGen, China), and the library was constructed using the NEB Next Ultra DNA Library Prep Kit. The library was quantified using q-PCR and sequenced on an Illumina sequencing platform.

For untargeted metabolomic analysis, 100 mg of each fecal sample was mixed with 500 µL of 80% methanol and incubated on ice for 5 minutes. The samples were then centrifuged at 15,000 rpm for 20 minutes at 4°C. The supernatant was diluted with LC-MS-grade water to reduce the methanol concentration to 53%, followed by a second round of centrifugation. The final supernatant was collected for further analysis. Metabolomic profiles were obtained using a Vanquish UHPLC system (Thermo Fisher, Germany) coupled to an Orbitrap Q Exactive HF-X mass spectrometer (Thermo Fisher, Germany). Chromatographic separation was performed on a Hypersil Gold column (100 × 2.1 mm, 1.9 µm) with a linear gradient for 17 min. Eluents used were as follows:

- Positive mode: eluent A (0.1% formic acid in water) and eluent B (methanol).
- Negative mode: eluent A (5 mM ammonium acetate, pH 9.0) and eluent B (methanol).

## Data analysis

Sequence data were analyzed using the Quantitative Insights into Microbial Ecology (QIIME 1.80) pipeline. Sequences were clustered into operational taxonomic units (OTUs) with ≥97% similarity using VSEARCH (v2.3.4) software. Alpha diversity indices (Chao1, observed features, and Faith's PD) were calculated to assess community richness and diversity, and statistical differences were evaluated using the Kruskal–Wallis test. Principal coordinate analysis (PCoA) was used to assess beta diversity. Linear discriminant analysis effect size (LEfSe) was performed to identify biomarkers by calculating linear discriminant analysis (LDA) scores (LDA ≥ 2, $P < 0.05$). Spearman's rank correlation analysis was used to evaluate relationships between serum biomarkers and fecal microbiota species, and correlation heat maps were generated.

Raw metabolomics data were converted to mzXML format using Proteowizard software. Peak identification, filtering, and alignment were performed using the XCMS package (R, v3.1.3). Metabolites were annotated by querying the KEGG, HMDB, and LIPIDMaps databases. For multivariate analysis, partial least squares discriminant analysis (PLS-DA) and orthogonal partial least squares discriminant analysis (OPLS-DA) were used to identify potential biomarkers. To validate the discriminatory power of these models, permutation tests were performed, assessing the statistical significance of the observed differences in metabolite profiles. Differential metabolites were selected based on the following criteria: VIP > 1, $P < 0.05$, and fold change ≥ 1.5. Metabolic pathways were enriched using the KEGG database. Pathways with $P < 0.05$ and a ratio of enriched metabolites ($x/n > y/n$) were considered to be significantly enriched. Receiver operating characteristic (ROC) curve analysis was used to evaluate the diagnostic performance of the metabolites.

## Statistical analysis

Data were analyzed using R (version 4.4.1) and GraphPad Prism (version 9.5). Statistical comparisons between two groups were performed using a two-tailed unpaired $t$-test, and differences among three groups were assessed using one-way ANOVA. The performance of the biomarkers was evaluated using ROC curves and AUC. The false discovery rate (FDR) method was applied for multiple comparisons in this study. Statistical significance was set at $P < 0.05$.

## RESULTS

### Clinical characteristics and laboratory parameters

Table 1 presents the clinical characteristics and laboratory parameters of the participants. There was no significant difference in the mean age between the pre-AET, post-AET, and HC groups (58.80 ± 6.65, 58.80 ± 6.65, and 54.1 ± 2.38 years, respectively; $P = 0.115$). All participants in the pre-AET, post-AET, and HC groups were women. Regarding clinical stage, pre-AET participants were mostly in the early stage (50% in stage II-A) and locally advanced stages (10% in stage II-A to III-A, 20% in stage III-A, and 20% in stage III-C). Most participants in both the pre-AET and post-AET groups were HER2-positive (60%), with Luminal B being the predominant molecular subtype (90%). The most common histological type in both groups was invasive ductal carcinoma (90%). Regarding the endocrine therapy in the post-AET group, 90% received aromatase inhibitors (letrozole and exemestane), and 10% received tamoxifen. Laboratory parameters, including lipid profiles (TG, TCH, HDL, LDL, ApoA1, and ApoB) and liver enzymes (ALT, AST, GGT, and CG), did not show significant differences between the pre-AET, post-AET, and HC groups (all $P > 0.05$). However, significant differences were observed in ALT and AST levels, which were significantly higher in both the pre-AET and post-AET groups than in the HC group (ALT: $P = 0.025$; AST: $P = 0.035$). Additionally, estradiol (E2), a hormone relevant to estrogen receptor-positive breast cancer, was significantly lower in the post-AET group ($P = 0.000$). The values for all blood indicators, including estradiol (E2), for each patient are provided in Table S1.

### Characterization of gut microbiota in breast cancer

The total number of operational taxonomic units (OTUs) identified in the pre-AET, post-AET, and HC groups was 1,749, 2,056, and 2,277, respectively. The HC group exhibited the highest number of unique features (1,715), whereas the pre-AET group exhibited the lowest number of unique features (1,242) (Fig. 1A). Alpha diversity analysis revealed significant differences in gut microbiota diversity among the pre-AET, post-AET, and HC groups. Species richness (Chao1) and phylogenetic diversity (Faith's PD) were significantly lower in the pre-AET group than in the HC group (Fig. 1B), indicating reduced microbial diversity in breast cancer patients before AET.

Beta diversity analysis (Fig. 1C) demonstrated distinct clustering of gut microbiota composition among the pre-AET, post-AET, and HC groups, highlighting notable differences in the microbial communities across these groups. At the phylum level (Fig. 1D, left panel), *Proteobacteria* and *Firmicutes_A* were the most dominant taxa in all groups. *Proteobacteria* were most abundant in the pre-AET group, whereas *Firmicutes_A* showed the highest relative abundance in the HC group. At the genus level (Fig. 1D, right panel), *Escherichia_710834* was more prevalent in the pre-AET group, whereas *Blautia_A_141781* was more abundant in the HC group. These results indicate distinct microbial profiles in patients with breast cancer before and after AET, with reduced microbial diversity observed in patients before treatment and notable shifts in microbial composition associated with therapy.

To identify differentially abundant gut microbiota among the pre-AET, post-AET, and HC groups, LEfSe analysis was performed at the genus level using a linear discriminant analysis (LDA) score of >2. Distinct microbial profiles were observed between the groups (Fig. 2). The HC group exhibited enrichment in 19 genera, including members of *Firmicutes* (e.g., *Roseburia*, *Limosilactobacillus*, and *Parvimonas*) and *Spirochaetota* (e.g., *Treponema_D* and *Treponemataceae*). The pre-AET group showed enrichment in nine genera, including *Bifidobacterium_388775*, *Coprococcus_A_187866*, and *Erysipelatoclostridium*. In contrast, the post-AET group demonstrated the highest diversity, with 31 enriched genera, including *Firmicutes* (*Dorea_A*, *Holdemanella*, and *Megasphaera_A_38685*) and *Proteobacteria* (e.g., *Methylobacterium*, *Proteus*, and *Cupriavidus*). These findings highlight substantial shifts in microbial composition post-AET, with

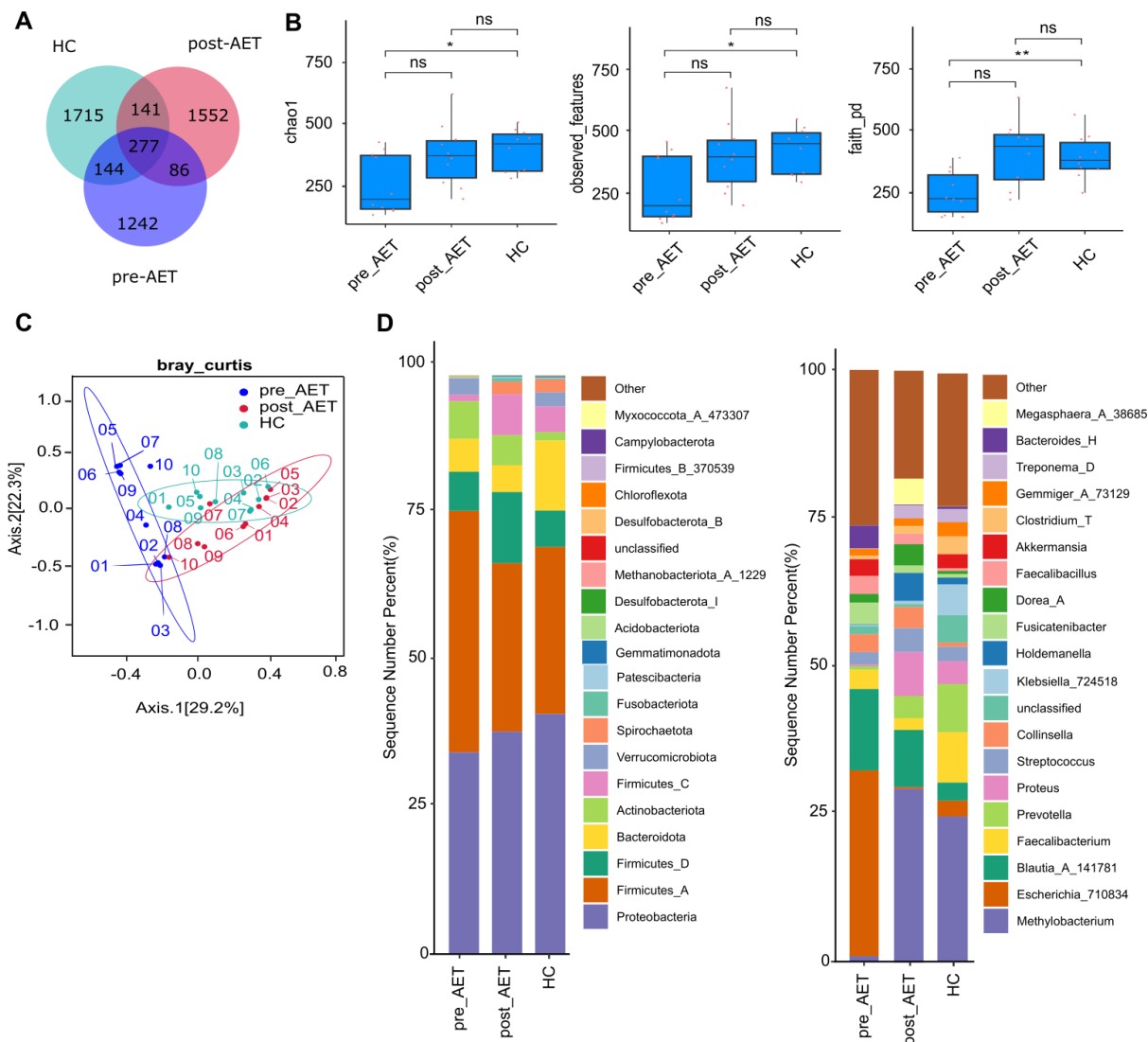

**FIG 1** Microbial diversity and community composition in patients with breast cancer. (A) Venn diagram showing the shared and unique microbial taxa among the pre-AET, post-AET, and HC groups, with overlapping areas representing taxa common to multiple groups and non-overlapping areas indicating unique taxa in each group. (B) Box plots of alpha diversity indices, including Chao1, observed features, and Faith's phylogenetic diversity, comparing microbial diversity across the three groups. Significant differences are marked with asterisks (*$P < 0.05$, **$P < 0.01$), and "ns" indicates non-significance. (C) Principal coordinate analysis (PCoA) plot based on Bray–Curtis dissimilarity, illustrating the clustering of microbial communities in the pre-AET, post-AET, and HC groups. Numbers 01–10 represent individual samples for each cohort. (D) Stacked bar charts representing the relative abundance of microbial phyla (left) and genera (right) in the pre-AET, post-AET, and HC groups. Different colors denote specific taxonomic groups, highlighting changes in microbial composition associated with adjuvant endocrine therapy (AET).

unique microbiota signatures observed in both the HC and post-AET groups and reduced diversity in the pre-AET group.

## Correlation analysis between gut microbiota and blood indicators

A correlation analysis was performed to evaluate the relationships between the gut microbiota at the genus level and various blood indicators, revealing significant associations (Fig. 3). CG levels were notably correlated with several genera. Among the positive correlations, *Achromobacter* exhibited the strongest association ($P < 0.001$), followed by *Bifidobacterium_388775* and *Halomonas_640386* ($P < 0.01$), *Ruminococcus_B*, and *Faecalicoccus* ($P < 0.05$). In contrast, significant negative correlations were observed with *Cupriavidus*, AG41, *Gemella*, *Longimicrobium*, *Shuttleworthia*, and *Methylobacterium* ($P <$

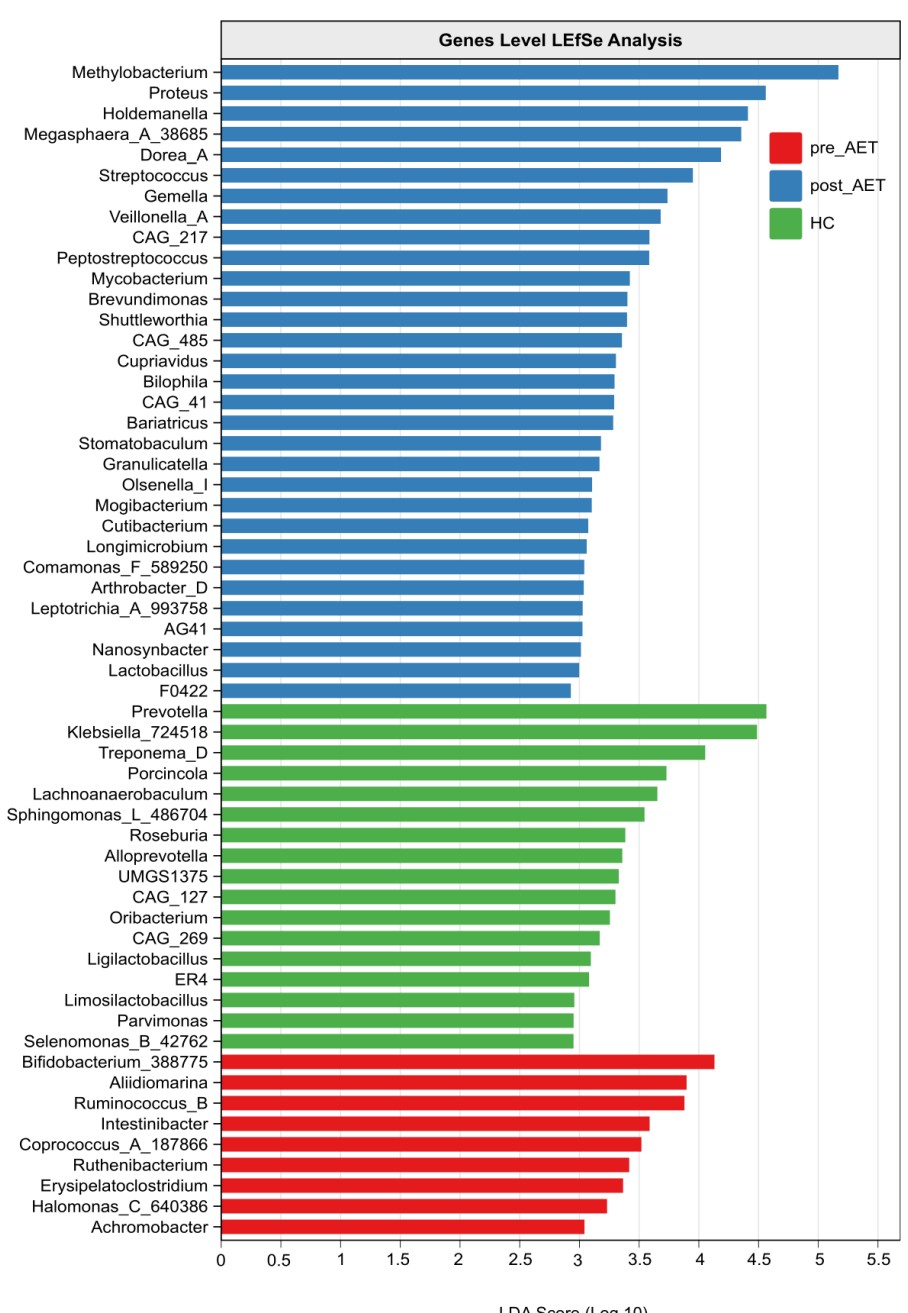

**FIG 2** LEfSe analysis of the gut microbiota at the genus level among the pre-AET, post-AET, and HC groups. Differentially abundant genera with an LDA score >2 are shown, with bars representing the effect size (LDA scores). Genera enriched in the pre-AET group are highlighted in red, post-AET in blue, and HC in green.

0.01), as well as F0422, *Peptostreptococcus*, *Cutibacterium*, *Pseudoalteromonas*, UMGS1375, and *Prevotella* ($P < 0.05$).

Estradiol (E2) levels were positively correlated with *the abundance of Peptoniphilus_C*. In contrast, E2 levels were significantly negatively correlated with several genera. Strong negative correlations were observed with *Peptostreptococcus*, *Cupriavidus*, and *Cutibacterium* ($P < 0.01$), whereas weaker negative correlations were observed with CAG_217, *Megasphaera_A_38565*, *Blautia_A_141780*, AG41, *Gemella*, *Longimicrobium*, *Shuttleworthia*, *Methylobacterium*, F0422, and *Prevotella* ($P < 0.05$). LDL levels were positively correlated with NSJ_61 ($P < 0.05$). Negative correlations were observed with CAG_217, *Faecalicoccus* ($P < 0.05$), *Megasphaera_A_38565* ($P < 0.01$), and CAG_317_143701 ($P <$

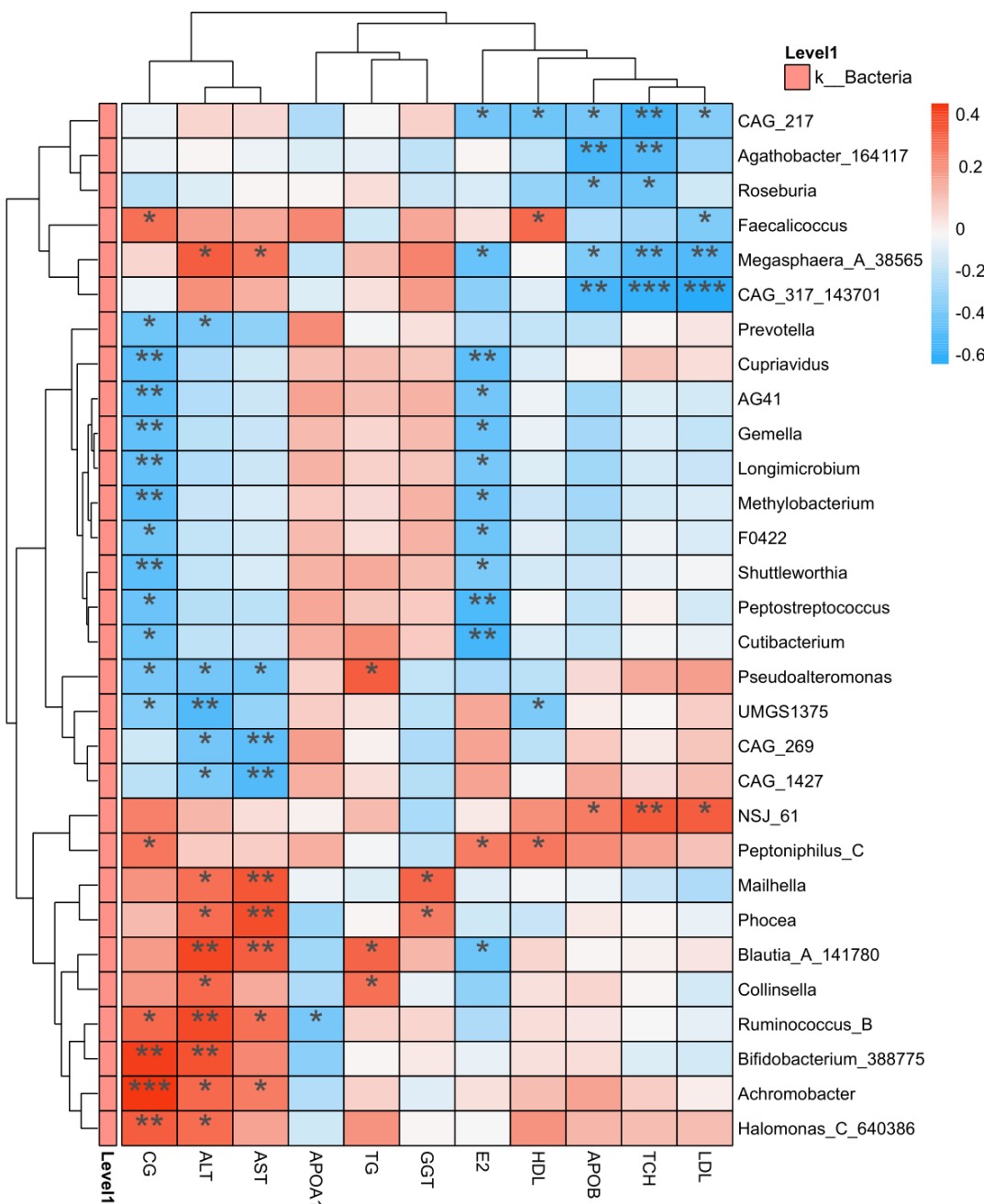

**FIG 3** Heatmap displaying the correlation between blood indicators (*X*-axis) and microbial species at the genus level (*Y*-axis). *R*-values (rank correlation) are represented by color, with red indicating positive correlations and blue indicating negative correlations. Darker colors represent stronger correlations between the variables.

0.001). HDL levels were positively correlated with *Faecalicoccus* and *Peptoniphilus_C* (*P* < 0.05) and negatively correlated with CAG_217 (*P* < 0.05). These findings highlight microbial taxa that may influence lipid metabolism. TG levels were positively correlated with *Collinsella*, *Blautia*, and *Pseudoalteromonas* (*P* < 0.05), but no significant negative correlations were observed.

ALT levels were significantly positively correlated with *Mailhella*, *Phocea*, *Collinsella*, *Megasphaera_A_38565*, *Achromobacter*, and *Halomonas_C_640386* (*P* < 0.05) and with *Blautia_A_141780*, *Ruminococcus_B*, and *Bifidobacterium_388775* (*P* < 0.01). Negative correlations were observed between CAG_269, CAG_1427, *Pseudoalteromonas*, and

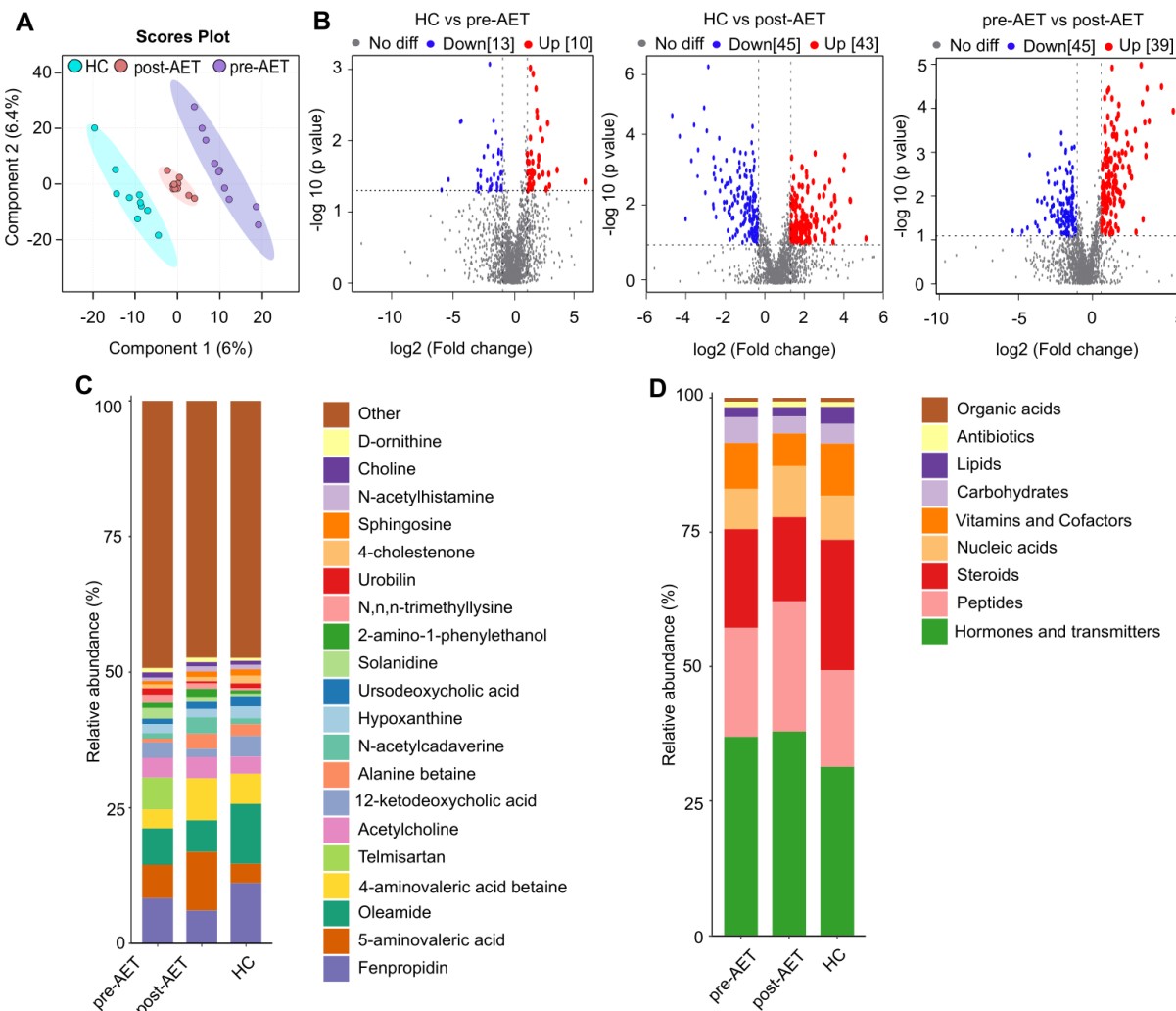

**FIG 4** Metabolomic analysis of gut microbial metabolites in the pre-AET, post-AET, and HC groups. (A) PLS-DA score plot depicting the separation of metabolite profiles among the healthy control (HC), pre-adjuvant endocrine therapy (pre-AET), and post-adjuvant endocrine therapy (post-AET) groups along components 1 and 2. The clear clustering patterns indicate distinct metabolite compositions across the three groups. (B) Volcano plots highlighting significantly altered metabolites between groups. (C) Stacked bar chart represents the relative abundance of the top 20 metabolites in the pre-AET, post-AET, and HC groups. (D) Stacked bar chart showing the distribution of metabolite categories (e.g., organic acids, lipids, and carbohydrates) across groups, highlighting shifts in metabolic classes.

*Prevotella* ($P < 0.05$) and UMGS1375 ($P < 0.01$). AST levels showed significant positive correlations with *Mailhella*, *Phocea*, and *Blautia_A_141780* ($P < 0.01$), as well as with *Achromobacter*, *Ruminococcus_B*, and *Megasphaera_A_38565* ($P < 0.05$). Negative correlations were observed between *Pseudoalteromonas* ($P < 0.05$), CAG_269, and CAG_1427 ($P < 0.01$). These findings indicate that specific microbial taxa may influence liver function through their association with ALT and AST levels. Gamma-glutamyl transferase (GGT) levels were positively correlated with *Mailhella* and *Phocea* ($P < 0.05$), with no significant negative correlations observed.

In summary, these results revealed significant correlations between the gut microbiota and blood indicators, suggesting complex interactions between host clinical parameters and microbial communities. These findings underscore the potential roles of specific gut microbes in modulating lipid metabolism, liver function, and hormonal regulation in our study population.

## Metabolite content analysis in breast cancer before and after therapy

To visually represent the differences in metabolites among groups, metabolomic analysis was conducted, highlighting significant variations in gut microbial metabolites in breast cancer patients before and after adjuvant endocrine therapy (AET) compared to those in healthy controls (HC). Figure 4A presents the PLS-DA score plot, showing clear separations among the HC, pre-AET, and post-AET groups, indicating distinct metabolite profiles. The PLS-DA permutation test results demonstrated that the observed test statistic was significantly different from random distribution, confirming that the model's discrimination effect was not due to chance (Fig. S1). Figure 4B shows the top 20 metabolites in the HC, post-AET, and pre-AET groups. The bar chart shows that certain metabolites exhibited different relative abundance levels across groups. For example, HC and post-AET tended to show higher metabolite levels than pre-AET, indicating a potential recovery or shift towards the HC metabolite profile after AET treatment.

Moreover, the OPLS-DA score plot showed the clear separation between the groups (Fig. S2A), and the permutation test (Q2 = 0.229, $P$ = 0.04) (Fig. S2B) further validated the predictive reliability of the model, confirming the observed differences in metabolite composition. These distinct clustering patterns underscore the profound impact of breast cancer and AET on gut microbial metabolite profiles. Our analysis categorized these metabolites, showing that the relative proportions of organic acids, lipids, carbohydrates, vitamins, and other metabolite classes varied significantly among the groups (Fig. 4C). For instance, hormones and transmitters were more prominent in the post-AET group, possibly indicating an adaptive or therapeutic effect of AET on gut microbial metabolites (Fig. 4D).

Pathway enrichment analysis identified significant alterations in key metabolic pathways among the pre-AET, post-AET, and HC groups, highlighting the impact of breast cancer and AET on systemic metabolism. Notable pathways included vitamin B6 metabolism, steroid hormone biosynthesis, sphingolipid metabolism, and amino acid metabolism, with vitamin B6 metabolism and steroid hormone biosynthesis showing the most significant enrichment (Fig. 5A). These findings suggest widespread disruptions in metabolic processes, including lipid, protein, and hormone metabolism. Metabolite abundance analysis revealed significant differences between the groups (Fig. 5B). Vitamin B6-related metabolites, including pyridoxine, pyridoxal, and 4-pyridoxate, were significantly higher in the HC group compared to pre-AET and post-AET groups, suggesting altered nutrient absorption or metabolic demands in cancer patients and during therapy. In contrast, cholesterol levels were markedly elevated in the pre-AET group, whereas the post-AET and HC groups showed reduced levels, indicating a potential normalization effect of AET on lipid metabolism. Similarly, key steroid metabolites, such as estradiol-17β, estriol, and estrone-3-sulfate, were significantly reduced in the post-AET group compared to the pre-AET group, reflecting the therapeutic effects of AET in suppressing estrogen biosynthesis. These results underscore the profound metabolic shifts associated with breast cancer and its treatment, with potential implications for modulating key pathways and metabolite profiles to restore systemic balance.

## Correlations between key microbial species and metabolite

The heatmap highlights the significant correlations between bacterial genera and metabolites across the three study groups (Fig. 6). Among the positive correlations, *Klebsiella_724518* showed statistically significant positive correlations with estradiol-17β ($P$ < 0.01) and pyridoxal and 4-pyridoxate ($P$ < 0.05), suggesting its potential role in influencing steroid hormone and vitamin B6 metabolism in the body. Similarly, *Proteus* exhibited a positive correlation with estrone-3-sulfate ($P$ < 0.01). Additionally, *Dorea_A* showed a significant positive correlation with pyridoxine ($P$ < 0.05), and *Prevotella* displayed a strong positive relationship with 4-pyridoxate.

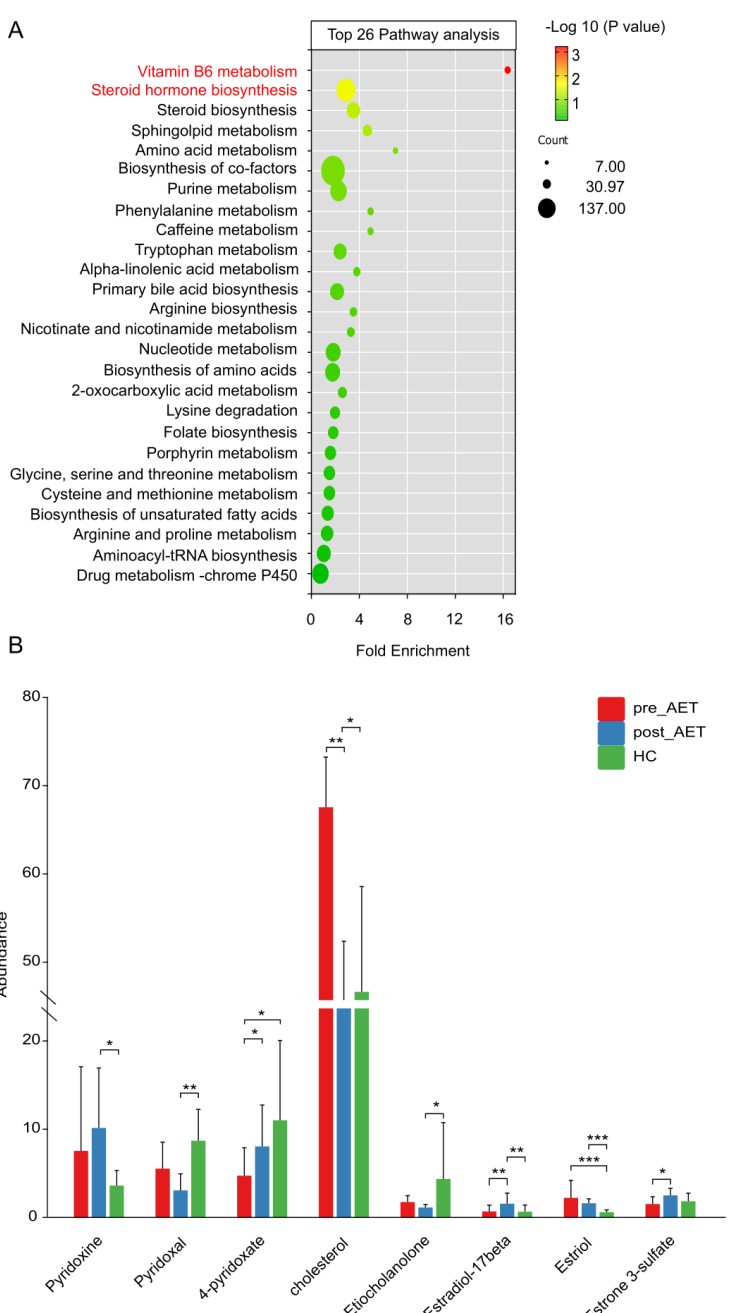

FIG 5 Pathway analysis and metabolite abundance in the pre-AET, post-AET, and HC groups. (A) Top 26 enriched metabolic pathways derived from the identified differential metabolites across the pre-AET, post-AET, and HC groups. The pathways were ranked by fold enrichment, with the bubble size indicating the number of metabolites involved and the color representing statistical significance ($-\log_{10}$ P-value). Notable pathways include vitamin B6 metabolism and steroid hormone biosynthesis, which showed significant enrichment. (B) Relative abundance of selected metabolites related to vitamin B6 and steroid hormone biosysnthesis. Significant differences (*P < 0.05, **P < 0.01, **P < 0.001) were observed across the groups for metabolites, such as pyridoxine, pyridoxal, and 4-pyridoxate (vitamin B6 metabolites), and steroid-related metabolites, including estradiol-17β, estriol, estrone-3-sulfate, and cholesterol. The pre-AET group showed elevated levels of cholesterol and certain steroid metabolites compared to the post-AET and HC groups, whereas vitamin B6 metabolites were more abundant in the HC group.

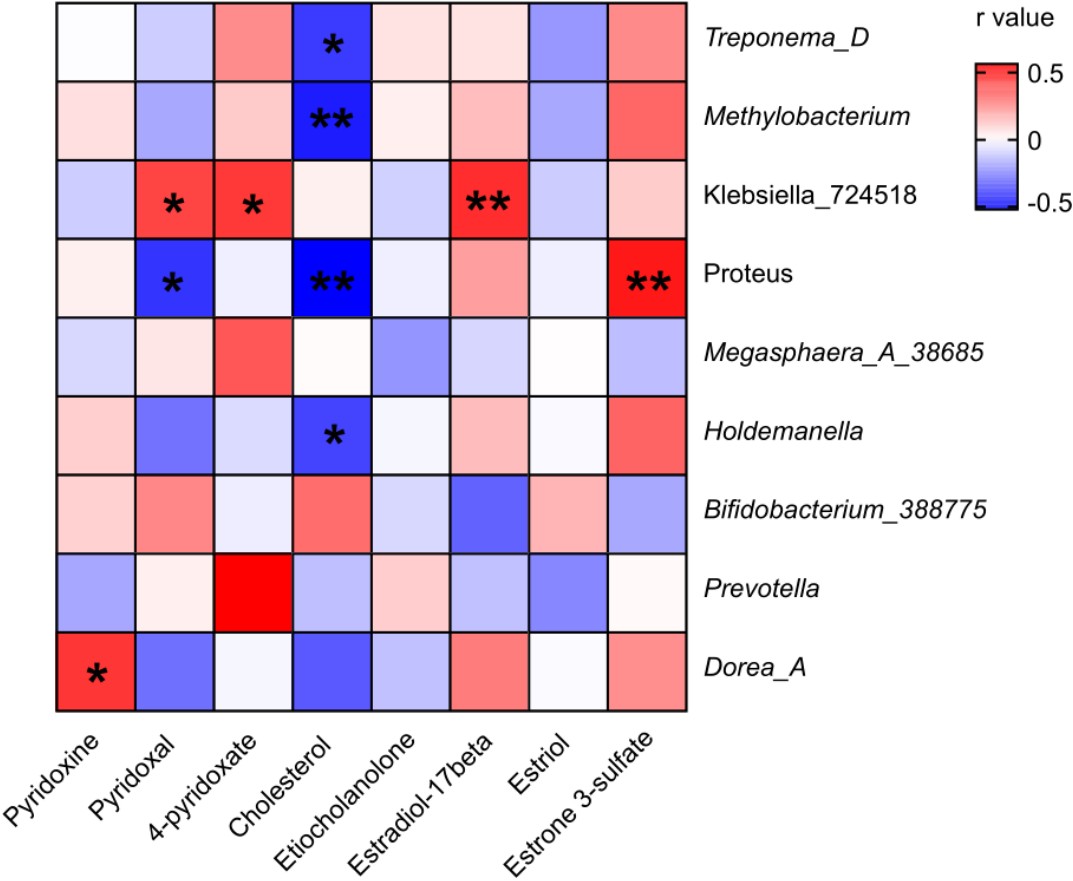

**FIG 6** Correlation heatmap between bacterial genera and selected metabolites. The heatmap shows the correlation coefficients (R-values) between the microbial genera (Y-axis) and metabolites (X-axis) across the different study groups. Red squares represent positive correlations, and blue squares indicate negative correlations, with color intensity reflecting the strength of the correlation. Statistical significance is indicated by asterisks: *$P < 0.05$, **$P < 0.01$.

In contrast, several bacterial genera exhibited significant negative correlations. *Proteus* showed strong negative correlations with pyridoxal ($P < 0.05$) and cholesterol ($P < 0.01$), potentially reflecting its impact on vitamin B6 and lipid metabolism. *Methylobacterium* ($P < 0.01$) and *Treponema_D* ($P < 0.05$) exhibited a strong negative correlation with cholesterol, suggesting an association with altered cholesterol metabolism. Similarly, *Holdemanella* also showed a negative relationship with cholesterol levels ($P < 0.05$). These findings suggest complex interactions between bacterial genera and host metabolites, particularly in pathways related to lipid and steroid metabolism.

## Diagnostic potential of metabolites for differentiating study groups

ROC curve analysis was conducted to assess the diagnostic potential of specific metabolites in distinguishing between healthy controls (HC), pre-AET, and post-AET groups. For the HC vs. pre-AET comparison, as shown in Fig. 7A, estriol demonstrates perfect discriminatory power with an AUC value of 1.000, making it the most effective biomarker for differentiating between healthy individuals and breast cancer patients before AET. Additionally, pyridoxal and 4-pyridoxate showed strong discriminatory abilities, with AUC values of 0.780, followed by cholesterol (AUC = 0.700). Other metabolites, such as pyridoxine (AUC = 0.560) and estradiol-17β (AUC = 0.520), showed weaker discriminatory performance.

For the pre-AET vs. post-AET comparison, as shown in Fig. 7B cholesterol (AUC = 0.880) and estradiol-17β (AUC = 0.870) emerge as the most effective biomarkers, reflecting significant metabolic changes after AET treatment. Estrone-3-sulfate

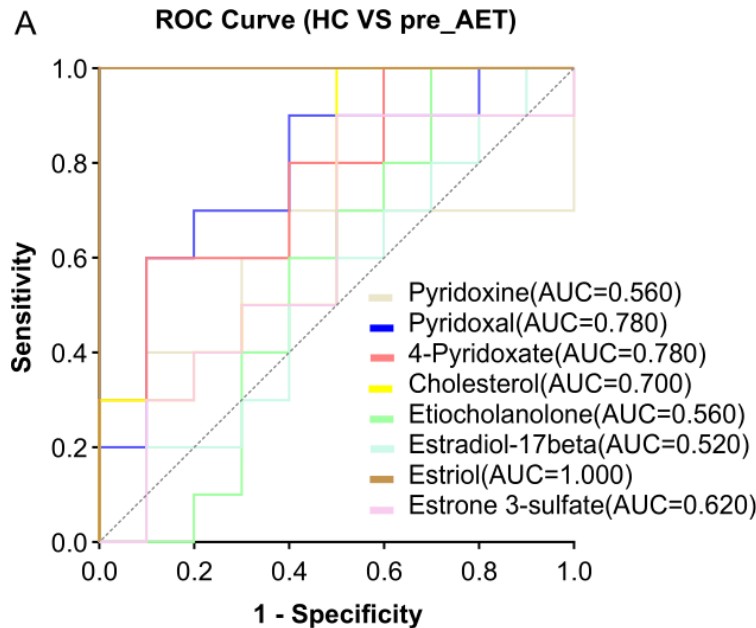

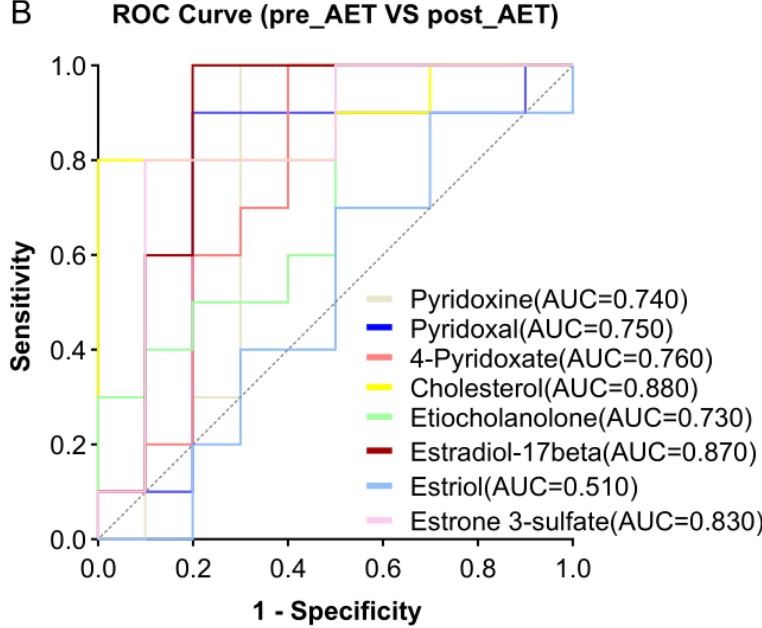

**FIG 7** Receiver operating characteristic (ROC) curves for discriminating metabolites between groups. (A) ROC curves for metabolites differentiating healthy controls (HC) from the pre-AET group, with area under the curve (AUC) values for each metabolite. Estriol shows the highest diagnostic performance (AUC = 1.000), followed by pyridoxal and 4-pyridoxate (AUC = 0.780). (B) ROC curves for metabolites distinguishing pre-AET from post-AET groups, with estradiol-17β showing the strongest discrimination (AUC = 0.870), followed by cholesterol (AUC = 0.880) and estrone-3-sulfate (AUC = 0.830). These results highlight the potential of these metabolites as biomarkers for disease progression and therapeutic response to AET.

(AUC = 0.830) and 4-pyridoxate (AUC = 0.760) also demonstrated strong discriminatory potential. Moderate performance was observed for pyridoxal (AUC = 0.750) and pyridoxine (AUC = 0.740). However, estriol showed limited discriminatory ability in this comparison (AUC = 0.510). These findings suggest that estriol, cholesterol, and estradiol-17β hold significant potential as biomarkers for distinguishing between

the study groups, with cholesterol and estradiol-17β being particularly relevant for monitoring the therapeutic response to AET.

## DISCUSSION

This study provides an in-depth exploration of the intricate interactions between the gut microbiota, metabolomic profiles, and effects of AET in patients with breast cancer. Our findings highlight significant alterations in both microbial composition and metabolite levels before and after AET compared to HCs, offering valuable insights into the role of the gut microbiota in modulating systemic metabolic pathways during cancer progression and treatment. We observed significant changes in gut microbiota and microbial function after AET, suggesting these communities may affect treatment outcomes and disease progression in hormone-sensitive breast cancer. In recent years, gut microbiota has been reported to be associated with various diseases (10, 27–31). Previous studies have reported the role of gut microbiota in cancer risk, including gastric (32), blood (28), lung (33), colorectal (34), and breast cancer (35). Our study showed significant differences in the gut microbial composition across the pre-AET, post-AET, and HC groups. The reduced alpha diversity in the pre-AET group compared to the HC group suggests that dysbiosis is linked to cancer progression, which is consistent with previous studies (35, 36). In the context of breast cancer, several factors can contribute to this dysbiosis, including tumor-induced inflammation, immune dysregulation, and changes in metabolic processes (24, 35). Regarding AET treatment, it is well known that drugs targeting the estrogen receptor pathway can influence the gut microbiota. While the tumor itself may contribute to microbial imbalance, the treatment may partially restore microbial diversity (24), as observed in our study. Moreover, LEfSe analysis at the genus level showed that the pre-AET group was less enriched than the control and post-AET groups, consistent with the previous study by (36), which showed an increased abundance of *Bifidobacterium* in cancer patients pre-AET. Moreover, we identified an increased abundance of *Coprococcus* and *Ruminococcus* in the pre-AET group. Both *Coprococcus* and *Ruminococcus* have been implicated in gut health and cancer regulation, particularly in the context of estrogen metabolism and anti-inflammatory effects (16, 37).

The clinical parameters of the study participants revealed no significant differences in age or sex distribution between the pre-AET, post-AET, and HC groups. However, as expected, patients with breast cancer exhibited varied clinical and pathological features, such as HER-2 expression. Interestingly, elevated liver markers (ALT and AST) in the pre-AET and post-AET groups could reflect systemic inflammation or therapy-induced hepatic stress. These results are consistent with those of previous studies showing that patients with breast cancer often experience liver dysfunction and metabolic dysregulation, especially during adjuvant therapies (38, 39). Correlation analysis between the gut microbiota and blood indicators revealed several significant associations, highlighting the complex interactions between microbial communities and systemic metabolic pathways. For example, estradiol (E2) levels were positively correlated with *Peptoniphilus_C*. In contrast, genera such as *Peptostreptococcus*, *Cupriavidus*, and *Cutibacterium* were negatively correlated with E2 levels. This suggests that these gut microbial species may play a role in modulating estrogen levels, which is crucial for breast cancer progression, particularly in estrogen receptor-positive cases (40). The influence of gut microbiota on hormone-sensitive cancers aligns with the growing evidence of the bidirectional relationship between microbial composition and systemic metabolic pathways (41).

Certain metabolites produced by the gut microbiota play a significant role in breast cancer development, with microbiota-metabolite interactions being crucial to the progression of the disease (25, 35). Our analysis identified correlations between specific genera and key metabolites, suggesting their influence on the host metabolism. Notably, *Klebsiella_724518* was positively correlated with estradiol, suggesting its involvement in estrogen metabolism. Previous studies have shown that gut microbiota species,

including *Bifidobacterium*, are implicated in estrogen metabolism (42). However, estradiol levels are dysregulated in breast cancer (16, 43). Pathway enrichment analysis revealed disruptions in vitamin B6 metabolism and steroid hormone biosynthesis, with reduced levels of metabolites, such as pyridoxine, pyridoxal, and 4-pyridoxate in the pre-AET and post-AET groups, suggesting metabolic stress and impaired nutrient processing. Regarding vitamin B6, some gut bacteria have been reported to metabolize B6 into its active form, pyridoxal-5'-phosphate (44), though the exact species involved in this process were not specifically analyzed in our study. Vitamin B6 plays a critical role in enzymatic reactions, immune function, and inflammation, and its deficiency may exacerbate metabolic imbalances (45–48). Elevated cholesterol and estrogen metabolite levels in the pre-AET group reflected increased lipid and steroid metabolism, whereas post-AET reductions in these metabolites, particularly estradiol-17β, estriol, and estrone-3-sulfate, confirm AET's efficacy of AET in suppressing estrogen biosynthesis. These findings align with the established mechanisms of AET, which targets estrogen signaling pathways to inhibit tumor progression in hormone-sensitive breast cancer (49, 50). Heatmap analysis further revealed that *Klebsiella_724518* was associated with estradiol-17β, pyridoxal, and 4-pyridoxate, whereas *Proteus* was correlated with estrone-3-sulfate. Conversely, *Methylobacterium* and *Treponema_D* were negatively correlated with cholesterol levels. These findings highlight the critical role of the gut microbiota in regulating key metabolic pathways disrupted in breast cancer and its treatment (43). Notably, our study focused on post-menopausal women with ER-positive breast cancer, and findings may not be directly applicable to younger populations, as significant differences in gut microbiota exist between pre-menopausal and post-menopausal individuals (6, 16). Additionally, since ER-positive breast cancer is estrogen-dependent, the results may not be generalizable to other types of breast cancer, although they may still provide valuable insights for future research.

ROC curve analysis provided insights into the diagnostic and therapeutic monitoring potentials of specific metabolites. In the comparison between the HC and pre-AET groups, estriol stood out as an exceptional diagnostic biomarker, achieving a perfect AUC of 1.00 and demonstrating its strong potential for identifying breast cancer before treatment initiation. These results were validated by permutation tests for the PLS-DA and OPLS-DA models as shown in Fig. S1 and S2B, which included estriol as a key biomarker, and the observed test statistics were significantly different from random distributions, confirming that the observed differences in metabolite profiles, including estriol, were not due to chance. Prior studies have linked elevated estrogen levels to breast cancer risk due to their proliferative effects on mammary tissue (51); however, few have specifically examined the discriminatory power of estriol. This positions estriol as a candidate for further validation in early detection panels, possibly complementing imaging or other liquid biopsy markers for early detection. However, its biological role in tumor initiation remains unclear, and future studies should explore whether estriol is a bystander or a driver of early carcinogenesis. Vitamin B6 metabolites (pyridoxal and 4-pyridoxate, AUC = 0.780) showed strong diagnostic performance, consistent with their significantly reduced levels in patients versus controls (Fig. 5B) reflecting disruptions in vitamin B6 metabolism. This depletion may reflect either increased metabolic demands during cancer progression or microbe-mediated vitamin B6 dysregulation (Fig. 6, as suggested by *Proteus* correlations). Vitamin B6 derivatives are critical for one-carbon metabolism, DNA synthesis, and immune modulation pathways, which are frequently dysregulated in cancer (52). Epidemiological studies have associated low vitamin B6 status with a higher risk of cancer (53). Clinically, these findings could inform adjuvant strategies, such as dietary B6 supplementation or monitoring B6 metabolite levels, to predict disease aggressiveness.

The robust performance of cholesterol (AUC = 0.880) and estradiol-17β (AUC = 0.870) in distinguishing pre- and post-AET patients highlights their utility in monitoring treatment efficacy. The importance of cholesterol in this context is biologically plausible, as it serves as a precursor for the synthesis of steroid hormones. AET, such

as aromatase inhibitors, directly inhibits estrogen production from androgens. Elevated levels of cholesterol have been associated with breast cancer progression, with preclinical studies demonstrating that tumor cells utilize cholesterol for the synthesis of membranes and signaling pathways (54). The notable decline in estradiol-17β following treatment further confirms the pharmacological effects of AET, aligning with findings from previous studies. Therefore, tracking estradiol and cholesterol levels over time can provide valuable clinical insights into breast cancer progression and treatment response. These metabolites could be leveraged in clinical practice to monitor adherence or detect early resistance, although their specificity to AET (vs. other therapies) requires further study. Furthermore, the reduced discriminatory ability of estriol (AUC = 0.510) post-AET suggests that it may be more relevant for diagnostic purposes than for monitoring treatment effects. This may reflect AET's selective suppression of estrogens (e.g., estradiol) over estriol, or the differential expression of enzymes involved in estriol metabolism (55). Alternatively, the diagnostic signal of estriol could arise from tumor-secreted factors altered early in carcinogenesis but unaffected by AET. This dichotomy underscores the need for context-specific biomarkers and cautions against extrapolating diagnostic utility for treatment tracking.

Overall, this study provides valuable insights; however, some limitations warrant further exploration. Given the exploratory nature of this study and the sample size larger, multicenter studies are needed to validate these findings. Future studies should explore the longitudinal changes in the microbiota and metabolites during AET to better understand the dynamics of these interactions. Moreover, mechanistic studies are required to elucidate the precise roles of different AET drugs and specific microbial taxa and metabolites in breast cancer pathophysiology. In conclusion, the integration of gut microbiota composition, metabolomic pathway analysis, and ROC curve-based biomarker discovery provides a comprehensive understanding of the metabolic disruptions associated with breast cancer and AET. Identifying key microbial genera, pathways, and metabolites offers opportunities to develop microbiome-targeted therapies and metabolite-based diagnostic tools. These findings pave the way for further research into the dynamic interplay between the gut microbiota and host metabolism, with implications for personalized breast cancer management.

## AUTHOR AFFILIATIONS

[1]Department of Cell Biology and Genetics, Shantou University Medical College, Shantou, China
[2]Department of Pharmacology, Shantou University Medical College, Shantou, China
[3]The Second Affiliated Hospital of Shantou University Medical College, Shantou, China
[4]Department of Pediatrics, Federal Medical College, Islamabad, Pakistan

## AUTHOR ORCIDs

Taha Majid Mahmood Sheikh  http://orcid.org/0000-0002-4385-9128
Muhammad Shafiq  https://orcid.org/0000-0002-4346-5903
Xiaoyang Jiao  http://orcid.org/0000-0002-7568-0595
Weitao Shen  http://orcid.org/0000-0002-1407-5797

## FUNDING

| Funder | Grant(s) | Author(s) |
|---|---|---|
| Guangdong Basic and Applied Basic Research Foundation | 2024A1515012460 | Xiaoyang Jiao |
| Shantou Medical Health Science and Technology Plan | 240429076498021 | Weitao Shen |
| Medical Scientific Research Foundation of Guangdong Province, China | B2022199 | Weitao Shen |

## AUTHOR CONTRIBUTIONS

Taha Majid Mahmood Sheikh, Formal analysis, Methodology, Software, Writing – original draft, Writing – review and editing | Fen Yao, Supervision | Zhenyan Liu, data curation, Formal analysis | Muhammad Shafiq, Writing – review and editing | Jilong Wu, data curation | Areeba Khalid, Software | Qingdong Xie, Conceptualization | Xiaoyang Jiao, Funding acquisition, Supervision, Writing – review and editing | Weitao Shen, Conceptualization, Funding acquisition, Methodology, Project administration, Resources, Supervision, Validation, Visualization

## DATA AVAILABILITY

Bacterial genomic sequences were deposited in the National Center for Biotechnology Information Sequence Read Archive (Accession Number: PRJNA1243283) available at https://www.ncbi.nlm.nih.gov/bioproject/PRJNA1243283. A STORMS (Strengthening the Organizing and Reporting of Microbiome Studies) checklist (T. M. M. Sheikh, 2025) is available at https://doi.org/10.5281/zenodo.15702367.

## ETHICS APPROVAL

Studies involving human participants were reviewed and approved by the Shantou University Medical College. Written informed consent for participation was obtained from all the participants for this study, in accordance with national legislation and institutional requirements. This study was approved by the Ethics Committee of the Second Affiliated Hospital of Shantou University (No. 2021-138), Guangdong Province, China.

## ADDITIONAL FILES

The following material is available online.

### Supplemental Material

**Figure S1 (mSystems00879-25-s0001.tif).** Permutation test results for PLS-DA model.
**Figure S2 (mSystems00879-25-s0002.tif).** OPLS-DA score plot and permutation test results.
**Table S1 (mSystems00879-25-s0003.xlsx).** Blood parameters of each participant.

### Open Peer Review

**PEER REVIEW HISTORY (review-history.pdf).** An accounting of the reviewer comments and feedback.

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
