## [Reviewer comments · mSystems]

Integrated Insights into Gut Microbiota and Metabolomic Landscape in Breast Cancer Patients Undergoing Adjuvant Endocrine Therapy

Taha Majid Mahmood Sheikh, Fen Yao, Zhenyan Liu, Muhammad Shafiq, Jilong Wu, Areeba Khalid, Qingdong Xie, Xiaoyang Jiao, and Shen Weitao

Corresponding Author(s): Taha Majid Mahmood Sheikh, Shantou University Medical College

Review Timeline:

Submission Date:

June 23, 2025

Accepted:

July 7, 2025

Editor: Hongwei Zhou

Reviewer(s): Disclosure of reviewer identity is with reference to reviewer comments included in decision letter(s). The following individuals involved in review of your submission have agreed to reveal their identity: Abid ullah shah (Reviewer #2)

Transaction Report:

DOI: <https://doi.org/10.1128/msystems.00879-25>

Re: mSystems00879-25 (**Integrated Insights into Gut Microbiota and Metabolomic Landscape in Breast Cancer Patients Undergoing Adjuvant Endocrine Therapy**)

Dear Dr. Taha Majid Mahmood Sheikh:

Your manuscript has been accepted, and I am forwarding it to the ASM production staff for publication. Your paper will first be checked to make sure all elements meet the technical requirements. ASM staff will contact you if anything needs to be revised before copyediting and production can begin. Otherwise, you will be notified when your proofs are ready to be viewed.

Sincerely,
Hongwei Zhou
Editor
mSystems